# Peroxisome Proliferator-Activated Receptors as Molecular Links between Caloric Restriction and Circadian Rhythm

**DOI:** 10.3390/nu12113476

**Published:** 2020-11-12

**Authors:** Kalina Duszka, Walter Wahli

**Affiliations:** 1Department of Nutritional Sciences, University of Vienna, 1090 Vienna, Austria; 2Center for Integrative Genomics, University of Lausanne, Le Génopode, CH-1015 Lausanne, Switzerland; Walter.Wahli@unil.ch; 3Lee Kong Chian School of Medicine, Nanyang Technological University Singapore, Clinical Sciences Building, 11 Mandalay Road, Singapore 308232, Singapore; 4Toxalim, INRAE, Chemin de Tournefeuille 180, F-31027 Toulouse, France

**Keywords:** caloric restriction, nuclear receptors, circadian rhythm, metabolism

## Abstract

The circadian rhythm plays a chief role in the adaptation of all bodily processes to internal and environmental changes on the daily basis. Next to light/dark phases, feeding patterns constitute the most essential element entraining daily oscillations, and therefore, timely and appropriate restrictive diets have a great capacity to restore the circadian rhythm. One of the restrictive nutritional approaches, caloric restriction (CR) achieves stunning results in extending health span and life span via coordinated changes in multiple biological functions from the molecular, cellular, to the whole–body levels. The main molecular pathways affected by CR include mTOR, insulin signaling, AMPK, and sirtuins. Members of the family of nuclear receptors, the three peroxisome proliferator–activated receptors (PPARs), PPARα, PPARβ/δ, and PPARγ take part in the modulation of these pathways. In this non-systematic review, we describe the molecular interconnection between circadian rhythm, CR–associated pathways, and PPARs. Further, we identify a link between circadian rhythm and the outcomes of CR on the whole–body level including oxidative stress, inflammation, and aging. Since PPARs contribute to many changes triggered by CR, we discuss the potential involvement of PPARs in bridging CR and circadian rhythm.

## 1. Introduction

The circadian (circa–approximately; dian–day) clock is a system conserved from cyanobacteria to humans, which generates internal oscillations with a near 24 h periodicity coordinating bodily processes in an anticipatory fashion. In mammals, the circadian clock governs nearly all aspects of physiology and behavior, including hepatic metabolism, cardiovascular activity, gastrointestinal tract activity, blood pressure, endocrine secretions, body temperature, renal function, and sleep–wake rhythm [1,2,3,4]. The adaptation to the recurring external and internal changes results in diurnal activity cycles and the anticipation of such changes constitutes a survival advantage to an organism capable of adjusting its molecular, physiological, and behavioral pathways [1,2,5].

The expression of circadian genes oscillates in an organ–dependent manner, with roughly 9–20% of all transcripts, which may differ from one cell or tissue to another, exhibiting a rhythmic pattern [6,7,8,9]. In fact, some studies report up to 43% of protein–coding genes showing circadian oscillations [10]. Disruption of the circadian coordination either by mutations in circadian clock genes or environmental factors interferes with metabolic processes and creates hormone imbalance, which increases the risk of several diseases, including various cancers, neurological and metabolic disorders, cardiovascular diseases, psychological and sleep disorders, all in all resulting in reduced longevity [2,11,12,13,14,15,16,17]. Forcing nocturnal animals to be active or to feed during the daytime, leads to obesity and metabolic disturbances [18,19]. Similarly, introducing a dim light in the dark phase shifts the time of food intake that results in dysmetabolism and increased body mass in wild–type (WT) mice [18]. In humans, circadian misalignment induces a reduction of daily energy expenditure by approximately 12–16% [20], decreases leptin and peptide YY levels which signal satiety, elevates postprandial glucose and insulin, and increases mean arterial pressure [21]. Metabolic and feeding cues act as potent zeitgebers for peripheral circadian clocks [5,22,23,24] and a large number of hepatic rhythmic transcripts under *ad libitum* feeding conditions are driven by the daily pattern in food intake. Nearly 10% of the hepatic transcriptome oscillations are under the control of the cell–autonomous circadian clock and feeding schedule. Out of nearly 3000 of hepatic transcripts oscillating daily, fasting suppresses the oscillation of >80% of them [25]. Refeeding after fasting triggers changes in the hepatic expression of the core circadian proteins including Period 1 and 2 (PER1, PER2), differentially expressed in chondrocytes 1 (DEC1 or BHLHE40), and REV–ERBα in rats and mice within 30 min to 1h [26,27]. Daytime feeding in naturally nocturnal mice leads to a shift in the phase of expression of >84% transcripts of the clock components and other rhythmic metabolic factors, and >73% of the oscillating genes increase their oscillation amplitude [25]. However, granting food access only in the nocturnal phase can induce WT–like anticipatory activity, respiratory exchange ratio (RER) changes, and gene expression rhythmicity in cryptochrome (*Cry*) deficient mice [25]. Therefore, besides the functional circadian clock, the availability of food, and the temporal pattern of feeding plays an essential role in determining daily rhythms. Similarly, time–restricted feeding (TRF) of mice promotes natural feeding rhythms and restores oscillations of the circadian clock and expression of their target genes, which rhythmicity is dampened by a high–fat diet (HFD) [28,29]. HFD is known to affect the central and peripheral circadian clocks; it disrupts circadian rhythms of the locomotor and feeding activities and leads to metabolic dysfunction due to the reduction of oscillation amplitude of the circadian clock and expression of the clock–controlled gene in the liver [30]. TRF besides restoring liver rhythms in circadian clock gene expression also normalizes metabolism and reduces obesity–associated inflammation without reducing food intake [28]. However, it is controversial whether similar metabolic benefits can be obtained without synchronization of the peripheral and central clocks namely when the mice are restricted to feed only during the light versus only the dark phase [29,31,32]. Similarly to animal studies, restriction of the eating window to day time in overweight people, without limiting energy intake, results in a reduction in body weight [33].

Therefore, diets, particularly restrictive diets that limit time or amount of consumption, have the capacity to adjust the expression of peripheral circadian genes. Resetting of circadian rhythms and introducing synchrony of the central and peripheral clocks can lead to increased longevity and wellbeing. Moreover, the circadian clock, as a vital factor in longevity, metabolism, and regulation of multiple physiological processes likely contributes to the beneficial effects of restrictive diets in mammals. In this non-systematic review, we will summarize the interconnection between one of the restrictive diets, caloric restriction (CR), and the circadian rhythm at the molecular and whole–organism level. Furthermore, we will present evidence for and potential of the nuclear receptors peroxisome proliferator–activated receptors (PPARs) in bridging CR and circadian processes.

## 2. Molecular Background of the Circadian Clock

In mammals, every tissue or physiological function exhibits diurnal oscillations. The central circadian pacemaker clock is located in the suprachiasmatic nucleus (SCN) of the brain. More precisely it resides in a bilateral group of neurons found in the anterior hypothalamus [2]. The master clock is entrained by light through the retinohypothalamic tract (RHT) whose role is to synchronize the internal clock ticking to the external day/night cycle. The perceived light leads to activation and synchronization of SCN neurons by an intrinsic SCN factor called vasoactive intestinal polypeptide (VIP) [34]. Then, these SCN neurons send signals to peripheral oscillators to prevent the dampening of circadian rhythms in various tissues. The signal distribution involves autonomic innervation or circulating humoral factors [2,35] such as transforming growth factor α (TGFα) [36], glucocorticoid hormones [37], prokineticin 2 (PK2) [38], and cardiotrophin–like cytokine (CLC) [39]. Additionally, the feeding/fasting schedule is an important external cue that contributes to the adjustment of the peripheral circadian clocks located in different organs [28,40].

At the molecular level, the circadian clock is driven by several transcription factors governing the core feedback loop including the transcriptional activators circadian locomotor output cycles kaput (CLOCK) [41,42] and brain and muscle ARNT–like protein 1 (BMAL1) as well as the transcriptional repressors PER1, PER2, and PER3 and CRY1 and CRY2 [43]. Structurally BMAL1 and CLOCK share one basic helix–loop–helix (BHLH) and two Per–Arnt–Sim (PAS) regions. These domains are necessary for heterodimerization and DNA binding. BMAL1 and CLOCK form a dimeric transcription factor that binds to E–box (5′–CACGTG–3′) and E–box–like promoter elements to activate transcription of the *Per* and *Cry* genes, as well as Rev–Erbs (*Rev–Erbα* and *β*), retinoic acid–related orphan receptors (Rors: *Rorα*, *β*, and *γ*) and other clock–controlled genes (Figure 1). In turn, PERs and CRYs heterodimerize, translocate to the cell nucleus and inhibit the CLOCK:BMAL1 complex [2]. The second feedback loop is comprised of REV–ERBs [44] and RORs [45], which bind to the ROR response element (RORE) in the *Bmal1* and *Cry1* promoter to rhythmically activate (REV–ERBs) or repress (RORs) transcription [46]. The alternating occupancy of RORE by RORs or REV–ERBα occurs as a result of the robust rhythmic fluctuations of REV–ERBα levels following enhanced transcription of the *Rev–erbα* gene by CLOCK:BMAL1 [44,45,47]. To achieve a nearly 24h cycle of the clock machinery, changes in the concentration of clock proteins and subcellular localization, post–transcriptional microRNA regulation, post–translational modifications including phosphorylation, acetylation, SUMOylation, methylation, and ubiquitination are crucial [48,49,50,51]. Circadian phosphorylation mediated by several kinases (GSK3β, AMPK, PKA, CAMK–1, CKIα, CKIδ, and CKIε), phosphatases (PP1, PP2A, PP4, and PP5) regulates localization and stability of integral core clock proteins [52].

## 3. Caloric Restriction

CR is a dietary approach involving a simple reduction of total energy intake and resulting in a myriad of beneficial responses ranging from the molecular cellular level to the whole–body system. CR increases mean and maximum life span in several species, including various strains of rats and mice, yeasts, worms, fruit flies, fishes, hamsters, owls, dogs, and cows [53,54,55,56,57]. In rodents, lifelong CR may extend lifespans by up to 50%, with weaker effects when CR is initiated later in life [58]. CR both delays the onset and slows–down the progression of several diseases, including age–related type 2 diabetes, sarcopenia, cancer, atrophy of the brain grey matter, and autoimmune, neurodegenerative, and cardiovascular diseases [53,54,55,56,57]. The metabolic adaptation to CR and the foundation of its beneficial effects originate from a synchronized response of a complex network of pathways, the most essential of them being under the control of insulin/insulin–like growth factor 1 (IGF–1), sirtuins (SIRTs), adenosine monophosphate (AMP) activated protein kinase (AMPK), and target of rapamycin (TOR). Additionally, multiple factors (among them thyroid hormones, adipokines, and ghrelin) and bodily systems (including sympathetic and neuroendocrine systems) also contribute to the beneficial outcomes of CR [53]. As a consequence of the multi–level coordinated response, CR impacts the whole body in terms of diminished inflammation, resting metabolic rate, body temperature, fat content, and enhanced insulin sensitivity [59].

## 4. The Impact of CR on Circadian Rhythm

Multiple studies investigated the crosstalk between CR and circadian rhythm. At the gene expression level, two clock genes, *Per1* and *Per2*, are upregulated by CR in several tissues [60]. Comparison of gene expression in seven different tissues of mice submitted to CR highlighted circadian rhythms with *Per2* being consistently upregulated in all of the tested tissues [61]. Besides *Per1* and Per2, CR increases the daily average expression of hepatic *Cry2* and *Bmal1* but reduces the protein level of Clock at every time points tested thorough the day [23]. The effect of CR on the expression of some of the core circadian genes is affected in the liver of *Bmal1* KO mice, which are considered as genetically arrhythmic. However, the induction of *Cry1* expression by CR is preserved in these *Bmal1* KO mice, suggesting the occurrence of both BMAL1–dependent as well as–independent regulatory mechanisms [23]. Of note, some aspects of circadian rhythms in behavior and physiology are sexually dimorphic [62,63]. The expression of the *Rev–Erbα*, *Rorγ*, *Cry1*, and *Cry2* genes along with several genes regulated by CR including flavin–containing monooxygenase *3* (*Fmo3*), major urinary protein 4 (*Mup4*), serpin family A member 12 (*Serpina12*), and cytochrome P450 4A12 (*Cyp4a12*) differs between males and females, whereas the liver expression of some circadian clock genes such as *Bmal1*, *Per1*, *Per2*, and *Per3* as well as the effect of CR on their expression are gender–independent [64].

Importantly, classical CR in a laboratory set up, when mice are given one portion of chow daily, resembles TRF in animal models, as hungry, restricted animals tend to consume all or most of their food portion within a short period of time after the delivery. Both, CR, and TRF can entrain the peripheral circadian clocks leading to the synchrony of metabolism and physiology. However, as opposed to TRF, CR entrains the clock in the SCN [65,66,67]. Moreover, CR–induced changes in hepatic gene expression are different from those of TRF, suggesting that CR exerts its effects by other means than periodic feeding [23].

The impact of CR on circadian rhythm has been evinced on multiple levels. In addition to peripheral effects, CR also resets the expression of several circadian clock genes in the SCN [68]. In addition, at the molecular level, most polysome–associated mRNA transcripts that are rhythmic in *ad libitum* fed mice lose their rhythms, and many new transcripts gain rhythms under CR. Only a small share of genes, including the circadian clock transcripts, preserve oscillations under CR as well as *ad libitum* feeding [69]. Besides gene expression, also global protein acetylation is highly circadian. Importantly, the acetylation rhythm is dampened in aged mice and can be rescued by CR [70]. Thus, CR regulates circadian clock gene expression in different tissues and affects both central and peripheral clocks at the transcriptional, translational, and posttranslational levels.

During CR, the organism adapts to the limited available resources and synchronizes its metabolic activities to produce metabolic enzymes efficiently to maintain homeostasis, relying on phases of energy supply. CR likely recruits the circadian clock for this metabolic optimization. Changes in nutritional availability of metabolic cues are implicated in lifespan extension and attenuation of the aging–related phenotype in a clock–dependent manner [22,71,72]. Due to the here mentioned multiple interlinks between CR and the circadian clock, implicating a large number of metabolic pathways under the control of the circadian clock, some of the beneficial outcomes of CR likely take place via clock regulation. The crosstalk between the circadian clock pathways participating in CR mechanisms, such as the sirtuin, insulin/IGF, and mammalian target of rapamycin (m)TOR signaling pathways will be described in the following chapters. Importantly, as recently reviewed [73], PPARs are implicated in every aspect of molecular signaling and outcomes of CR.

## 5. Molecular Pathways Affected by CR and Their Implication in the Circadian Rhythm

### 5.1. Insulin Signaling in Metabolism and Circadian Rhythm Regulation

Insulin/IGF–1 pathway activity plays a major role in the control of aging [74,75,76,77,78] and in the beneficial effects of CR [79,80]. Insulin is a key regulator of glucose uptake and utilization in insulin–responsive tissues. Following food intake, increased blood glucose levels trigger pancreatic β–cells to secrete insulin. Free circulating insulin activates insulin receptors on the surface of target cells eliciting a signaling cascade initiated by the activation of insulin receptor substrates (IRS 1–4) followed by phosphorylation of phosphoinositide 3–kinase (PI3K), which manages metabolic response including PDK1 and Akt stimulation by phosphorylation. Akt signaling prompts glucose transporter 4 (GLUT4) to translocate to the cell membrane where it initiates cellular glucose uptake. Akt also stimulates the production of glycogen and inhibits gluconeogenesis. Moreover, Akt activates mTOR, which facilitates anabolic processes, while mTORC2 feeds back to regulate Akt [81]. Insulin signaling affects multiple downstream pathways including mitogen–activated protein kinase (MAPK), which controls growth, sterol regulatory element–binding protein 1 (SREBP–1), which stimulates the synthesis of lipid and cholesterol as well as the family of Forkhead (FOXO) transcription factors regulating metabolism and autophagy [82,83]. Inhibition of IGF–1/PI3K/Akt signaling contributes to the anti–cancer and DNA–repair activity of CR [84,85,86].

The direct interconnection between CR and circadian rhythms has been evinced by the fact that *Igf–1* expression is regulated by both CR and the circadian clock [64]. Interestingly, *Igf–1* expression is rhythmic and shows sexual dimorphism [64]. Furthermore, a genome–wide RNAi screen for genes that regulate cellular clock functions in human cells identified the insulin signaling pathway as the most overrepresented pathway [87]. Accordingly, the impact of CR on plasma IGF–1 and insulin level is compromised in mice deficient for *Bmal1* [71]. Further, the postprandial release of insulin resets peripheral clocks by regulating the expression of core circadian genes. Insulin rapidly increases the expression of *Per2 in* insulin–sensitive tissues like the liver, muscle, or adipose tissue, but not the lung or brain [88]. Insulin secreted upon refeeding–after–fasting stimulates *Per2* and reduces *Rev–erbα* expression in hepatocytes [27]. The capacity of insulin to initiate entrainment of the liver clock was demonstrated by the administration of insulin in cultured rat hepatocytes which acutely induced expression of *Per1*, *Per2*, and *Dec1* [88]. Accordingly, inhibition of pathways downstream of insulin signaling, such as MAPK and PI3K, blocks the induction of the *Per1* and *Per2* clock genes [88].

Next to insulin, glucose is a circadian clock regulator. Glucose levels control AMPK activity, which phosphorylates and controls the stability of the CRY proteins [89]. In rats, glucose infusion during the light phase strongly induces expression of *Per2* in the SCN and reduces *Per2* expression in the liver [90]. Accordingly, a reduced level of glucose availability delays the light–induced phase shift [91]. Glucose also reduces the expression of *Per1* and *Per2* in fibroblasts in vitro [92].

Reciprocally, the circadian clock in the pancreas regulates insulin and glucagon production and secretion and their signaling in SCN via the autonomic nervous system [93,94,95,96]. Glucagon secretion is also regulated by Rev–Erb*α* [94]. In vitro islet β–cells exhibit robust rhythms of both *Bmal1* and *Per1* [95,97], while disruption of circadian clock functions in pancreas–specific *Bmal1* KO mice leads to glucose intolerance, defective insulin production and secretion [95]. BMAL1 and CLOCK contribute to the regulation of the recovery from the hypoglycemic response to insulin [98]. Mice deficient in BMAL1 and CLOCK exhibit dysregulated glucose homeostasis, impaired glucose tolerance, and reduced insulin sensitivity [93]. On the contrary, KO of the negative arm of the circadian machinery, that is invalidation of *Crys*, *Pers, or*
*Rev–erbα* leads to increased insulin levels [99,100,101]. In the *Per2* KO mouse, insulin secretion is more effectively stimulated by glucose and its analogs compared to WT animals. At the same time, the circadian rhythm of hepatic insulin–degrading enzyme (Ide) is disrupted leading to decreased insulin clearance. Consequently, *Per2* KO animals suffer from hyperglycemia [100].

Control of glucose homeostasis requires both the central and peripheral clocks, and disruption of synchronization between them affects glucose metabolism negatively [102]. CLOCK drives the transcriptional stimulation of glycogen synthase 2 (*Gys–2*) and therefore modulates the circadian rhythms of hepatic glycogen synthesis [103]. BMAL1 and CLOCK stimulate gluconeogenesis that consequently is reduced in the KO models of these proteins [98]. Similarly, RORα directly induces phosphoenolpyruvate–carboxykinase (*Pepck*) expression [104] and thus, RORα deficiency, as well as treatment with RORα antagonists, inhibits PEPCK expression and glucose production [105]. Accordingly, glucose–6–phosphatase (G6Pase) and PEPCK are suppressed in HepG2 cells overexpressing *Rev–erbα*, encoding the physiological repressor of RORα. Accordingly, silencing *Rev–erbα* significantly increases the expression of G6Pase and PEPCK [104,106,107]. Alike, during fasting, rhythmically expressed CRY proteins in the liver reduce gluconeogenesis through phosphorylation of cAMP response element–binding protein (CREB) [108], and phosphoenolpyruvate carboxykinase 1 (PCK1) by direct interaction of CRY with the *Pck1* promoter [109]. Furthermore, during feeding and acute fasting, PER2 dampers gluconeogenesis and enhances glycogen storage by decreasing the activity of glycogen phosphorylase (GP) [110].

### 5.2. mTOR Signaling and the Circadian Rhythm

The mTOR pathway is a key effector pathway of CR and it is known for monitoring the availability of nutrients and regulating longevity. TOR is one of the Ser/Thr protein kinases from the family of phosphatidylinositol 3 (PI3) kinase–related kinases [111,112] and it functions as a key component of two complexes, mTORC1 and mTORC2 [111]. mTORC1 is sensitive to cellular energy levels, nutrient status, mitogenic signals, and oxygen levels and it is inhibited by rapamycin. mTORC1 signaling leads to the regulation of mRNA translation and autophagy. mTORC2 is not rapamycin sensitive and operates as a regulator of the cellular actin cytoskeleton [113,114]. The mTOR pathway integrates intracellular and extracellular physiological stimuli. In this pathway, the protein complex of tuberous sclerosis proteins 1 and 2 (TSC1 and TSC2) mediates upstream signals from growth factors, such as insulin and IGF–1 to mTORC1. Activation of mTOR mediates the phosphorylation of several executor proteins involved in mRNA translation and ribosome biogenesis, such as ribosomal S6 kinase 1 (S6K1) and eukaryotic initiation factor 4E–binding protein (4E–BP) [115,116,117]. mTORC1 downstream signaling controls autophagy and metabolism, including the glycolytic turnover and anabolic processes associated with the fed state including lipogenesis [118,119,120,121], cholesterol synthesis via activation of SREBP–1/2 [118,122,123], and protein synthesis [124,125].

mTOR activity in vivo is induced by the abundance of nutrients and gradually decreases during fasting. However, food–independent rhythmicity in activity and expression of the mTORC1 complex members has been observed in the SCN and liver, cardiac and skeletal muscles, adipocytes, and retinal photoreceptors but not in the intestine or lung [126,127,128,129,130,131,132,133,134]. In the mouse brain, mTORC1 activities exhibit daily alterations in the arcuate nucleus, hippocampus, and the frontal cortex [130,135,136], all regions that manage circadian rhythms, feeding, learning, memory, and emotions. The oscillations of mTOR activity are delimited by internal and external regulators [137]. In the brain of *Drosophila*, TOR rhythms are found particularly in the ventral lateral neurons [138,139]. Neuronal TORC1 and AKT signaling have been shown to drive behavior [138]. Also, in the SCN, mTORC1 signaling is activated by light and controls behavior in a circadian manner [126,127,140]. Brief light exposure of mice during the night, but not during the day, triggers instant phosphorylation of the mTOR translation effectors S6K1, S6 ribosomal protein (S6), and translational repressor 4E–BP1 [126]. A KO of 4E–BP1 in mice leads to a higher amplitude of molecular rhythms in the SCN, increased capacity for re–entrainment to a shifted light/dark cycle, and higher resistance to the disruption of rhythm by constant light [140]. In vivo, infusion of the inhibitor of mTOR1 rapamycin leads to an attenuation of the phase–delaying effect of early–night light. Equally, disruption of mTOR during the late–night augments the phase–advancing effect of light. At both the early– and late–night time points, abrogation of mTOR signaling leads to a significant attenuation of light–induced PER protein expression [141]. Conversely, constitutive activation of mTOR in *Tsc2*–deficient fibroblasts alters the dynamics of clock gene rhythmicity and raises levels of principal clock proteins, including CRY1, BMAL1, and CLOCK [142]. Heterozygous *mTor* KO mice present a lengthened circadian period of locomotor activity rhythms both in constant darkness and constant light [142]. mTOR inhibition lengthens the period and dampens the amplitude of circadian clock proteins, whereas mTOR activation shortens the period and augments amplitude in hepatocytes, adipocytes, and human U2OS cells [142,143]. In *Drosophila*, TOR modulates the circadian period in opposite direction compared to mice. Overexpressing S6K in the ventral lateral neurons, the central pacemaker cells, extends the circadian period [138]. Consistently, KO of *Tor* in Per expressing cells shortens the period of locomotor rhythms in *Drosophila* [139].

Importantly, the intracellular concentration of Mg^2+^ ions act as a cell–autonomous timekeeping component that controls key clock properties from unicellular algae to human. Furthermore, Mg^2+^ ions oscillations regulate circadian control of translation by mTOR [143].

In the liver, fasting/feeding cycles regulate the energy sensor AMPK and mTORC1 [28,129,130] independently of CRY1 and CRY2 but partly relying on BMAL1 [137]. S6K1 rhythmically phosphorylates BMAL1 at an evolutionarily conserved site to activate it for modulation of mRNA translation and protein synthesis. Therefore, protein synthesis rates show circadian oscillations that are dependent on BMAL1 [131], which deficiency increases mTORC1 activity both in vivo and in vitro [130]. Increased mTOR signaling correlates with faster aging while treatment with rapamycin extends the lifespan of prematurely–aging *Bmal1 KO* mice by 50% [130]. Therefore, increased BMAL1 activity likely contributes to lower mTORC1 activity in CR. In summary, the regulation of mTOR downstream and upstream of the circadian rhythm connects daily oscillations with sensing of the nutrient status (Figure 2).

### 5.3. AMPK as an Energy Sensor and Executor of Circadian Metabolic Activities

AMPK serves as a key energy sensor of the AMP:ATP and ADP:ATP ratios. AMPK, activated at low levels of ATP, suppresses energy–consuming anabolic pathways and promotes catabolism resulting in the recruitment of stored energy to reestablish the ATP levels [144]. AMPK impacts metabolism via several downstream effectors including CREB–regulated transcriptional coactivator–2 (CRTC2) [145], TBC1D1/AS160 [146,147], peroxisome proliferator–activated receptor γ coactivator 1 α (PGC–1α) [148], and histone deacetylase 5 (HDAC5) [149]. Functionally, AMPK inhibits fatty acid (FA), TG and cholesterol synthesis, while stimulating FA uptake and β–oxidation [150,151,152,153,154]. Furthermore, AMPK reduces protein synthesis by inhibiting mTOR [155]. AMPK also affects the metabolism of glucose by stimulating glycolysis [156] and inhibiting glycogen synthesis [157] and gluconeogenesis [158,159,160]. It also drives nutrient–induced insulin secretion from islet β–cells [161] and glucose uptake by GLUT4 [162]. Besides metabolism, AMPK plays a role in inflammation, cell growth, autophagy, and apoptosis [163].

The circadian clock is paired with cellular metabolic fluctuations via nutrient–sensing pathways. While the expression of six of the transcripts encoding subunits of mammalian AMPK does not vary during the day, that of the AMPKβ2 subunit is eight times higher in the middle of the day than it is at night. AMPKβ2 determines the cellular localization of the heterotrimeric AMPK complex [164]; therefore, the presence of the AMPKα1 subunit in the nucleus increases at the time of day when AMPKβ2 expression peaks [89]. The rhythmic hepatic activity and nuclear localization of AMPK correlate inversely with CRY1 levels, which is likely due to the fact that stimulation of AMPK leads to phosphorylation and degradation of CRY [89]. Further, activation of AMPK by 5–aminoimidazole–4–carboxamide ribonucleoside (AICAR) or metformin in mouse livers causes a phase shift of the clock, and animals in which the AMPK pathway is genetically disrupted show alterations in peripheral clocks [89,165]. Moreover, acute AICAR stimulation alters the expression of clock genes in WT mice but not in mice lacking the AMPKγ3 regulatory subunit [166]. Genetic disruption of either AMPKα1 or AMPKα2 subunit dampens the rhythm of body temperature, the free–running activity, changes the circadian pattern of core clock gene expression in mice in an isoform– and tissue–specific manners [165]. In mouse muscles, AMPK regulates the expression patterns of the circadian genes *Cry2*, *Rev–erbα*, and *Bhlhb2* (basic helix–loop–helix domain containing class B 2) [166]. Moreover, AMPK is capable of phosphorylating casein kinase ε (CKIε) and thereby increases its enzymatic activity, indirectly leading to a destabilization of PER2 [167]. PGC–1α, which co–activates the RORs and consequently stimulates the expression of *Bmal1* and *Rev–erbα,* is phosphorylated by AMPK [148,168,169]. PGC–1α is required for cell–autonomous clock function [169] and PGC–1α KO mice show an abnormal diurnal rhythm of physical activity, body temperature, and metabolic rate, due to disrupted expression of clock genes and genes involved in energy metabolism. Besides the direct impact on ATP levels, AMPK affects the energy status by promoting feeding through signaling in the hypothalamus as well as by adjusting circadian metabolism [170,171]. AMPK may thereby mediate the influence of fasting/feeding cycles on the circadian clock (Figure 2).

### 5.4. SIRT Energy Sensors in the Context of CR and Daily Rhythmicity

SIRTs serve as energy sensors by detecting the ratio of reduced to oxidized nicotinamide adenine dinucleotide NAD^+^:NADH and react as transcriptional effectors mostly through their HDAC activity. SIRTs are class III HDACs that manage processes connected with nucleic acid biology including DNA repair, homologous recombination, and histone deacetylation, as well as transcriptional gene silencing [172,173]. There are seven subtypes of SIRTs (SIRT1–7) in mice and humans which differ in their cellular localization and function. SIRT1–SIRT3, SIRT5, SIRT6, and SIRT7 act as deacetylases, while SIRT4 and SIRT6 have ADP–ribosylation activity. Besides histones, SIRTs modify also several transcriptional regulators including the nuclear factor kappa–light–chain enhancer of activated B cells (NF–κB), p53, FOXO, PGC–1α, as well as enzymes, including acetyl coenzyme A (CoA) synthetase 2 (AceCS2), long–chain acyl–coenzyme A dehydrogenase (LCAD), 3–hydroxy–3–methylglutaryl–CoA synthase 2 (HMGCS2), superoxide dismutase 2 (SOD–2), and structural proteins, such as α–tubulin [174,175,176,177,178]. Therefore, SIRTs impacts multiple processes and pathways including circadian clocks, cell cycle, mitochondrial biogenesis, and energy homeostasis, consequently influencing aging, apoptosis, inflammation, and stress resistance [179,180]. SIRT1 is mostly associated with metabolism. In *S. cerevisiae* an extra copy of the Sir2 gene, a yeast homolog of mammalian Sirt1, increases lifespan [181,182], whereas the deletion of Sir2 shortens it [181]. In yeast and *Drosophila*, lack of the Sirt1 homolog offsets CR–triggered life extension [183,184,185]. A yeast analog of Sirt1 takes part in DNA repair and regulates aging–related gene expression [186].

PGC–1α, one of the main metabolic effectors of SIRT1, is activated by SIRT1–mediated deacetylation [187,188]. Activated PGC–1α enhances hepatic gluconeogenesis [187], mitochondrial activity in muscle and BAT leading to increased exercise capacity and thermogenesis; consequently, PGC–1α promotes protection against obesity and metabolic dysfunction [189]. SIRTs interact also with factors involved in response to CR including the FOXO family of transcription factors [190,191,192], which affects gluconeogenesis and glucose release from hepatocytes [193], cell differentiation, metabolism as well as longevity regulation [194,195,196]. Further, AMPK enhances SIRT1 activity by increasing cellular NAD^+^ levels [197,198] and activation of SIRT1 may cause AMPK phosphorylation via deacetylation–dependent activation of the AMPK–activating kinase liver kinase B1 (LKB1) [199,200].

SIRT1 controls the circadian expression of the core clock genes *Bmal1*, *Rorγ*, *Per2*, and *Cry1*. Also, SIRT1 is recruited by CLOCK:BMAL1 chromatin at circadian promoters [201]. Further, the *l*evels of NAD^+^, NADP^+^, NADH, and NADPH affect the binding capacity of CLOCK–BMAL1 heterodimers to E–box elements [202]. Similarly, resveratrol, a polyphenolic SIRT activator, regulates the expression of clock genes *Per1*, *Per2*, and *Bmal1* in Rat–1 fibroblast cells [203]. It also modifies the rhythmic expression of clock genes (*Clock*, *Bmal1*, and *Per2*) and lipid metabolism–related genes controlled by the clock (*Pparα*, *Srebp–1c*, *Acc1*, and *Fas*) in HFD–fed mice [204] and reverses the change induced by high–fat feeding in the expression of Rev–Erbα in adipose tissue of rats [205]. SIRT1 also promotes the deacetylation and subsequent degradation of PER2 in a circadian manner [201].

The rhythmic acetylation of BMAL1 and acetyl–histone H3 Lys9/Lys14 at circadian promoters correlates with SIRT1 HDAC activity that is regulated in a circadian manner. Therefore, genetic or pharmacological inhibition of SIRT1 activity causes disturbances in the acetylation of H3 and BMAL1 and the circadian rhythm [206]. Moreover, the circadian transcription factor CLOCK has histone acetyltransferase (HAT) activity, and SIRT1 HDAC activity counteracts the HAT activity of CLOCK [201,206]. Another circadian protein, REV–ERBα regulates pancreatic glucagon secretion via the AMPK/nicotinamide phosphoribosyltransferase (NAMPT)/SIRT1 pathway [94]. Further, the expression of NAMPT, a rate–limiting enzyme involved in NAD^+^ production through the salvage pathway, is regulated by the BMAL/CLOCK heterodimer. Consequently, NAD^+^ levels exhibit rhythmic daily oscillations [207,208]. Moreover, by being recruited to the *Nampt* promoter, SIRT1 contributes to the circadian synthesis of its own coenzyme [208]. Inhibition of NAMPT promotes fluctuations of *Per2* by releasing CLOCK:BMAL1 from suppression by SIRT1. In turn, CLOCK binds to the *Nampt* promoter and stimulates its activity, thereby contributing to a feedback loop comprising NAMPT/NAD^+^ and SIRT1/CLOCK:BMAL1 [207].

SIRT3 also interacts with the circadian rhythm. It sets the pace in the acetylation and activity of oxidative enzymes and consequently respiration in isolated mitochondria. *Bmal1* KO mice have significantly decreased SIRT3 activity, which affects mitochondrial oxidative function, and supplementation with nicotinamide mononucleotide (NMN), a NAD^+^ precursor, restores SIRT3 function and enhances oxygen consumption in these animals [209]. Importantly, the rhythm of cyclic global protein acetylation dampens with aging in mice [70]. CR regulates SIRT1 activity and therefore it modulates the circadian acetylation of AceCS1, a pathway controlling rhythmic nucleocytoplasmic acetyl–CoA production [210,211]. Consequently, CR rescues the hepatic protein acetylation rhythm over the day/night cycle in mice. Accordingly, the circadian transcriptome of CR–mediated effects on circadian reprogramming and SIRT1–specific transcriptome overlaps [70] indicating a pivotal role of SIRT1 in connecting CR and the circadian rhythm (Figure 2).

## 6. PPARs

The ligand–dependent receptors PPARs form a nuclear receptor subfamily which comprises three isotypes, PPARα (NR1C1), PPARβ/δ (NR1C2), and PPARγ (NR1C3) [212,213,214]. PPARs play roles in multiple processes including FA and eicosanoid signaling, cell proliferation and differentiation, tissue repair and remodeling, bone formation, insulin sensitivity, glucose and lipid metabolism. Synthetic ligands [thiazolidinediones (TZD) and fibrates] of PPARs are commonly used in the therapy of glucose, and lipid disorders as well as in the prevention and treatment of cardiovascular and metabolic diseases [215,216,217]. Diverse FAs, phospholipids, prostacyclins, prostaglandins, and leukotrienes serve as natural ligands of PPARs [218,219] and also are involved in other biological processes, which indicate a potential link between PPARs and nutrition, metabolism, and inflammation. In response to CR, PPAR expression patterns vary vastly in different tissues and organs. In brief, the expression of PPARα, –β/δ, and –γ in the heart, PPARα in white adipose tissue (WAT), and PPARγ in the liver remains unchanged during CR. On the contrary, the expression of PPARβ/δ in the liver, PPARα in the spleen, and PPARα, –β/δ, and –γ in muscle is decreased, whereas that of PPARα in the liver and intestinal epithelium is increased [220,221,222,223,224,225,226,227,228]. In rat kidneys, 60% CR prevents age–related reduction in PPAR activity [229].

PPARs contribute to CR–related outcomes by regulating many pathways, all of which we have reviewed recently [73], such as mTOR, insulin signaling, AMPK, SIRT, inflammation, oxidative stress, mitochondrial function, energy metabolism, aging, hunger, and microbiota composition. Simultaneously, PPARs act as molecular connectors joining clock genes and specific rhythmic metabolic outputs. All PPARs exert their functions in a circadian manner and their expression is regulated by both core clock genes and clock–controlled genes. Moreover, PPARs and the core clock genes cross–regulate each other at the transcription level. Additionally, CLOCK/BMAL1 and PPARs synergistically regulate gene expression as CLOCK/BMAL1 heterodimers increase the transcriptional activity of some PPAR target genes. Upon PPAR response element (PPRE) removal from the promoter of these genes, CLOCK/BMAL1 loses its transcriptional effects [230]. Finally, it is important to bear in mind that there are multiple interactions between each PPAR isotype and the circadian rhythm (Figure 3).

### 6.1. PPARα

Most studies on the implication of PPARs in CR and circadian rhythm are about PPARα, which is particularly important for lipid metabolism and is expressed at relatively high levels in organs with significant FA catabolism, such as the liver and brown adipose tissue (BAT) [231]. PPARα synchronizes metabolism by regulating genes associated with FA transport, peroxisomal and mitochondrial β–oxidation, ketogenesis, and gluconeogenesis [218]. Considering the adaptational role of PPAR*α*, its ligands can be seen as mimetics of CR [232]. As a sensor of the energy status, PPARα adjusts metabolic processes to both, scarcity and abundance of nutrition.

During fasting, upon depletion of dietary glucose, PPARα enhances glucose import, glycolysis, and glycogenolysis [233,234,235,236]. It also increases FA oxidation and glucose–induced insulin secretion in β–cells [237,238], thereby improving glucose management [239], and insulin sensitivity [240,241,242,243,244]. Not surprisingly, PPARα KO mice show reduced expression of some of the genes involved in gluconeogenesis and glycogen metabolism [245] and they develop age–dependent hyperglycemia [246]. Following 24 h fasting, the PPARα KO mice display severe hypoglycemia together with elevated plasma insulin concentrations [247,248].

Fasting results in carbohydrate depletion, fat mobilization, and ketone body production, all of which rely on PPARs. In response to glucose shortage, PPARα supports FA uptake, β–oxidation, FA catabolism, lipogenesis, and ketone body production [248,249,250] by regulating the expression of most of the rate–limiting enzymes of ketogenesis, such as HMG–CoA synthase and of β–oxidation including ACOX1, enoyl–CoA hydratase, and 3–hydroxyacyl CoA dehydrogenase (EHHADH), carnitine palmitoyltransferases 1 and 2 (CPT1/2), medium, long and very long–chain acyl–CoA dehydrogenases (MCAD, LCAD, VLCAD), fibroblast growth factor 21 (FGF21). In the liver, 19% of genes whose expression is changed by CR and are involved in lipid metabolism, inflammation, and cell growth depends on PPAR*α* and is not affected by CR in PPAR*α* KO mice [232]. Consequently, PPARα KO mice show impaired fasting–induced hepatic responses, which causes hypoketogenesis and liver steatosis [248,249,250].

PPARα modulates an important aspect of CR response, hunger perception. Endocannabinoid oleoylethanolamide (OEA), a ligand of PPARα is particularly well recognized for appetite regulation [251]. OEA is synthesized in enterocytes in response to fat consumption and is increased by sympathetic activity and bile acids that activate N-acylphosphatidylethanolamines hydrolyzing phospholipase D (NAPE-PLD) [252,253]. OEA has an anorectic effect via PPARα activation and it reduces meal size or delays meal intake, which results in body weight loss [251,254,255,256]. OEA acts peripherally via afferent sensory fibers of the vagal nerve in the intestine. This signaling promotes the secretion of the neuropeptide oxytocin and satiety by increasing the expression of c-fos in the nucleus solitary tract of the brainstem and the paraventricular nucleus of the hypothalamus [257,258]. OEA synthesis is increased following gastric bypass surgery that requires intestinal PPARα expression to reduce food intake [259]. Moreover, a single nucleotide polymorphism (SNP) in PPARα is associated with an increased risk of high altitude appetite loss [260], and activation of hypothalamic PPARα normalizes food intake in hypophagic *Fas* KO mice [261]. Unexpectedly, challenging hepatic *Pparα* KO mice with carbohydrates revealed a PPARα-dependent glucose response for FGF21. This effect is associated with an increased preference of these mice for sucrose [262]. Finally, dietary protein restriction increases FGF21 expression in human and rodent plasma, an effect which depends on increased phosphorylation of eukaryotic initiation factor 2α (eIF2α) in the liver, eIF2α kinase general control nonderepressible 2 (GCN2), and PPARα signaling [263].

#### 6.1.1. PPARα and Molecular Effectors of CR

PPARs interact with all of the major pathways managing the response to CR. The diverse interactions between mTOR and PPARs impact FA synthesis, glucose metabolism, oncogenesis, and the immune system. mTORC1 adjusts ketone body production upon fasting [264] and mice with constitutive hepatic activation of mTORC1 present low PPARα levels and are not able to induce ketogenesis in response to fasting [264]. In the fed state, the insulin–dependent PI3K pathway activates mTORC1, which inhibits PPARα activity and ketogenesis [264]. Additionally, insulin–dependent phosphorylation and thus inhibition of cytoplasmic FA synthase (FAS), which is mediated by mTORC1, limits the synthesis of PPARα ligands. Conversely, when mTOR is inhibited during fasting, FAS generates endogenous PPARα ligands [265]. In the intestine, mTOR activity increases with age, which is accompanied by inhibition of PPARα. Consequently, levels of the primary Wnt feedback inhibitor, Notum increase, and Wnt signaling is reduced leading to the diminished regenerative capacity of stem cells in the Paneth cell niche [266]. Similarly, in the livers of old mice, elevated mTORC1 activity [264] coincides with diminished activity of PPARα and reduced ketogenesis [267,268,269]. Unsurprisingly, mTORC1 inhibition prevents aging–induced loss of PPARα activity and ketone body production [264]. Finally, CR induces autophagy in the liver and other organs by integrating, mTOR, and PPARα functions [270,271].

As already mentioned, AMPK and PPARα both sense the energy status and adjust metabolism to nutritional changes. PPARα stimulates the expression of proteins involved in restoring ATP levels and therefore regulates AMPK activity. Moreover, PPARα and AMPK act to regulate long–term and short–term FA oxidation, respectively. Therefore, during fasting, as glucose levels drop and FA levels rise, AMPK and PPARα coordinate their actions. High intracellular AMP levels induce AMPK, which stimulates mitochondrial FA uptake while PPARα enhances the hepatic β–oxidation capacity [218,265,272,273]. The activation of PPARα by AMPK occurs in several tissues including muscle, liver, and pancreas [274,275,276,277,278,279,280]. Agonist activation of PPARα also activates AMPK showing that the stimulation of AMPK and PPARα is reciprocal [281,282,283].

Similarly, there is a positive activation loop between PPARα and SIRT1. SIRT1 deletion in hepatocytes decreases PPARα activity and reduces FA β–oxidation, while SIRT1 overexpression stimulates PPARα target genes. Further, activation of PGC–1α by deacetylation, requires interaction between SIRT1 and PPARα [284]. Moreover, PPARα and SIRT1 mutually suppress the expression of genes regulating mitochondrial activities that are controlled by the estrogen–related receptors (ERRs) during fasting [285,286,287].

#### 6.1.2. PPARα in Circadian Rhythm

PPARα is expressed according to a diurnal rhythm in mouse liver, heart, kidney, and, to a smaller extent, in the SCN [288,289]. It is directly linked with the circadian clocks as it maintains the circadian oscillation of the *Bmal1* gene in the brain and muscle by binding to its promoter (Figure 3). Accordingly, fenofibrate, a PPARα ligand, promotes circadian clock gene expression in cell culture, up–regulates hepatic *Bmal1* in vivo, and *Pparα* KO mice exhibit altered hepatic circadian expression of *Bmal1* and *Per3* [290]. Reversely, BMAL1 is an upstream regulator of *Pparα* expression [290]. Similarly, the CLOCK–BMAL1 heterodimer transactivates *Pparα*. CLOCK interacts with the E–box–rich region of the *Pparα* promoter and, in homozygous *Clock* KO mice, the hepatic circadian expression of *Pparα* is abolished [291]. Interestingly, *Rev–erbα* is also a PPARα target gene [292].

Besides its transcription regulatory role, PPARα also interacts directly with clock proteins. PER2 interacts with PPARα and serves as its transcriptional coregulator. Consequently, PER2 is rhythmically bound at the promoters of PPAR target genes in vivo [293]. Considering multiple interactions between PPARα and circadian clock proteins it is important to reflect on the impact of medical drugs, mostly lipid–lowering ones, acting as PPARα ligands. Fenofibrate increases transcription and resets rhythmic expression of BMAL1, PER1, PER3, and REV–ERB*α* in mouse livers [290] and cultured hepatocytes [292]. Bezafibrate was shown to phase–advance the circadian expression of BMAL1, PER2, and REV–ERB*α* in the cortex, liver, and fat of mice but without affecting the SCN [294]. Additionally, bezafibrate advances the active phase of mice under light–dark conditions in a photoperiod–dependent manner; therefore, it is involved in the entrainment of the circadian clock to environmental light–dark conditions [294,295].

Lipid homeostasis is controlled by the circadian clock and disruption of rhythmicity results in dyslipidemia and obesity in various clock mutant mouse models. *Rev–Erbα* KO mice exhibit elevated VLDL triglyceride levels [296], in *Bmal* KO mice triglycerides oscillations are dampened in plasma [98], *Clock* KO mice show hyperlipidemia and hepatic steatosis [297] whereas *Per2* KO mice have reduced whole–body fat, total triglycerides as well as FAs in plasma [298]. Accordingly, around 15% of total metabolites undergo diurnal regulation in human plasma with 75% of them being lipids [299]. Similarly, about 13% of lipid metabolites exhibit circadian changes in human plasma [300], with a high prevalence of diglycerides, and triglycerides peaking in the middle of the day. Consistently, the oscillation of triglycerides in the rat plasma corresponds to the feeding cycle [301]. In mouse liver, 17% of quantified lipids display circadian rhythmicity with all triacylglycerols (TAG) reaching their peak levels around CT8 which parallels PPARα peak expression in the liver. Surprisingly, *Per1/2* KO mice show variations of a similar fraction of lipids, however, with a different composition than in the WT mice. Moreover, PPARα retains its circadian expression upon *Per1/2* deletion [302]. In the same way, adipose tissue (BAT, iWAT, and eWAT) shows a cyclic expression of the circadian oscillator genes including *Npas2*, *Bmal1*, *Per1–3*, and *Cry1–2*, in combination with *Pparα* fluctuations [303].

Consistent with its role in the circadian regulation of lipid metabolism, PPARα controls numerous genes implicated in lipid and cholesterol metabolism, and energy homeostasis. PPARα–driven cardiac regulation of enzymes involved in FA β–oxidation in the heart ensures metabolic homeostasis and heart performance [304,305]. The expression of FGF21 playing important roles in adaptation to fasting, such as lipolysis and ketogenesis is under the circadian influence of PPARα [306]. Many of the metabolic genes, such as *Srepb*, *Fas*, and β–hydroxy β–methylglutaryl–CoA reductase (*Hmg–CoAR*), display daily fluctuations in mouse liver. The rhythmical expression of *Fas* and *Hmg–CoA* over the day presents a pattern that is opposite of that of *Pparα*. In the *Pparα* KO mice, the rhythmical expression of lipogenic and cholesterogenic genes is dampened or eradicated. However, compared to WT mice, these changes are observed only during the active phase when food intake is high [307].

PPARα is considered one of the marker genes of CR. Its expression is altered in CR in multiple organs and it likely mediates the response to metabolic changes as well as metabolic switch to low–energy conditions including mobilization of energy stores. PPARα is also inseparably entwined in circadian clock control. Importantly, CR significantly enhances the circadian activity of PPARα and, thus the oscillatory expression of its target genes [69]. Consequently, PPARα is one of the best documented examples of a factor bridging CR and circadian rhythm.

### 6.2. PPARβ/δ

In contrast to the expression pattern of PPARα and PPARγ, PPARβ/δ is more ubiquitously expressed, with particularly high levels in several regions of the brain, kidneys, liver, skin, and gastrointestinal tract [308,309]. It is involved in cell proliferation, differentiation, tissue repair, placenta, and gut development, as well as energy homeostasis especially in muscle [280,310,311,312,313,314,315,316]. PPARβ/δ stimulates energy expenditure by enhancing FA oxidation in skeletal muscle and adipose tissue, thereby protecting against diet–induced obesity and insulin resistance [317,318,319,320]. Moreover, hepatic deletion of PPARβ/δ reduces, while its activation increases, FA uptake in muscles, pointing at coordinated production and utilization of lipids in functionally interacting organs [321]. Exercise–induced glycogen exhaustion triggers PPARβ/δ activity in rat muscle stimulating changes in energy sources [322]. PPARβ/δ is particularly well known for increasing exercise capacity by promoting the switch to type I muscle fibers, which stimulates mitochondrial activity and fat oxidation resulting in improved endurance [323,324]. Furthermore, PPARβ/δ modulates feeding responses and its neuronal deletion leads to increased susceptibility to diet-induced obesity, elevated fat mass and decreased lean mass on low-fat diet, accompanied by abnormal responses to fasting [325].

#### 6.2.1. PPARβ/δ and Molecular Effectors of CR

Only a few studies have documented a connection between mTOR and PPARβ/δ thus far. PPARβ/δ may impact mTOR activity by controlling FA metabolism and the production of phosphatidic acid, which directly activates the mTOR complex [326]. In non–small cell lung carcinoma (NSCLC) cells, the expression of PPARβ/δ is activated by nicotine through PI3K/mTOR [327], whereas GW501516, a PPARβ/δ agonist stimulates cell proliferation by inhibiting the expression of phosphatase and tensin homolog deleted on chromosome 10 (PTEN), which signals downstream to mTOR [328], picturing an interplay between the two pathways.

In models of metabolic syndrome [329], diabetes [330], and in overweight subjects [331], the GW501516 agonist of PPARβ/δ lowers plasma glucose and/or insulin levels. One of the mechanisms of energy status sensing by PPARβ/δ involves activation of cytosolic phospholipase A2 (cPLA_2_) by glucose surplus, which leads to hydrolysis and peroxidation of arachidonic and linoleic acid, producing PPARβ/δ ligands [332]. Activated PPARβ/δ represses the β–cell mass and insulin secretion [333]. It prevents insulin resistance in adipocytes [334], human liver cells [335], and skeletal muscle cells [335,336]. Joint complementary effects of PPARβ/δ in distinct tissues involving its impact on carbohydrate catabolism in the liver and β–oxidation in muscle results in improved metabolic homeostasis and insulin sensitivity [337].

AMPK and PPARβ/δ interact directly in the muscle to enhance exercise performance [338]. AICAR, an activator of AMPK, increases endurance, but the combination of AICAR and GW0742, an agonist of PPARβ/δ, significantly boosts the activity of the receptor and rises all running parameters. The mechanism behind involves a lowering in carbohydrate utilization with a parallel shift to fat as the main energy source in fatigued muscles [339]. Therefore, agonists of AMPK and PPARβ/δ are considered exercise mimetics [340].

Finally, PPARβ/δ increases transcription [341] as well as protein levels [342] of SIRT1. The regulation of SIRT1 and PPARβ/δ activity operates bidirectionally [342,343]. Moreover, PPARα and PPARβ/δ stimulate SIRT1–dependent osteogenic differentiation [344,345], whereas PPARγ prevents it [346].

#### 6.2.2. PPARβ/δ in Circadian Rhythm

In mice, the expression of *Pparβ/δ* mRNA oscillates daily in brown adipose tissue and muscle [288]. PPARβ/δ regulates the diurnal expression of lipogenic genes including acetyl–CoA carboxylase 1 and 2 (*Acc1* and *Acc2*), *Fas*, and stearoyl–CoA desaturase–1 (*Scd*–*1*), during mouse feeding cycle [321]. Moreover, hepatic *Pparβ/δ* is a target of mir–122 and likely mediates the impact of mir–122 on cholesterol and lipid metabolism. Importantly, mir–122 transcription is rhythmically regulated by REV–ERBα [347]. Finally, PPARβ/δ expression rhythmically changes in hamster SCN where it modulates glutamate release. Treatment with the PPARβ/δ agonist L–16504 enhances the phase delay of the locomotor response induced by a light pulse. This observation suggests that PPARβ/δ is mediating entrainment of the circadian clock by light (Figure 3) [348].

### 6.3. PPARγ

PPARγ is primarily known as a master regulator of adipogenesis, a target of insulin–sensitizing, and anticancerogenic therapies [349,350,351]. It has been shown to impact several types of cancers by reducing cell proliferation [352,353,354], stimulating cell differentiation [352,355], triggering apoptosis [352,353,356], and inhibiting angiogenesis [357]. PPARγ contributes to several processes including adipogenesis and FA storage, inflammation, and lipid and glucose metabolism [349,358,359], long–chain FA processing in the intestinal epithelium [360], and control of adiposity via the sympathetic nervous system [361]. Classical full PPARγ agonists TZDs such as rosiglitazone (Avandia) and pioglitazone (Actos) are used to be applied in the therapy of metabolic diseases [362]. Remarkably, dual PPARα/γ agonists have beneficial effects on both lipid and glucose metabolism. In addition to their antidiabetic properties, they also are hypolipemic, anti–inflammatory, hypotensive, and antiatherogenic, and exert anticoagulant effects [217,363,364,365]. Multiple natural compounds stimulate PPARγ expression and activity. They include nutrients, most importantly FAs and their metabolites. Furthermore, glutamine, curcumin, capsaicin, ginsenosides, and vitamin E impact PPARγ activity indirectly [366]. Importantly, bacterial metabolites and bacterial by–products [367,368,369], and specific bacterial strains in the gut microbiota also increase the expression and activity of PPARγ [368,370,371]. A common side effect of treatment with TZDs is weight gain. HFD-related weight gain partly depends on the effect of neuronal PPARγ signaling which leads to limited thermogenesis and increased food intake [372,373]. Moreover, PPARγ plays role in regulating food intake by stimulating adipogenesis since WAT secretes endocrine and paracrine satiety mediators like leptin, adiponectin, and resistin [374].

#### 6.3.1. PPARγ and Molecular Effectors of CR

Of all PPARs, the interplay of PPARγ with CR pathways has resulted in the richest set of evidence for an important role of PPARγ in the outcomes of CR. For instance, genetic variation in the *Pparγ* gene and its target gene *Acsl5* are implicated in the extend of weight reduction upon CR [375]. In particular, six *Pparγ* SNPs correlate with weight loss under CR [376]. The Pro(12) Ala substitution of *Pparγ* reduces the binding affinity for PPRE and therefore, this PPARγ variant has a weaker transcriptional activity [377,378]. The *Pparγ* (12)Ala allele confers resistance to CR-induced weight loss in obese women [375] whereas, following CR, female homozygotes for Ala(12)Ala regain more body weight than Pro (12) Pro homozygotes [379]. Further, *Pparγ* polymorphism is linked with BMI changes due to fat intake [380,381], diet FA composition [382], and plasma TG response to ω3–FA supplementation [383].

PPARγ is well known as a regulator of insulin sensitivity and TZD drugs ameliorate insulin resistance by reducing plasma glucose and they improve the lipidemic profile in type 2 diabetes (T2D) and in obese subjects without diabetes [384]. TZDs also sustain the function of pancreatic β–cell thereby reducing T2D incidence [385,386]. In addition to TZDs, several other PPARγ agonists including FMOC–L–leucine (F–L–Leu) [387], INT131 besylate [388], quercetin, and kaempferol [389,390] improve insulin and glucose management. PPARγ exerts its insulin–sensitizing effects in several ways. It promotes functional WAT necessary for proper glucose homeostasis as evinced by insulin resistance upon both partial and generalized lipodystrophies [391,392] as well as by improved glycemic control following PPARγ–triggered fat mass generation [393]. Moreover, the magnitude of PPARγ–elicited insulin sensitization is proportional to the reduction of lipid accumulation in skeletal muscle [394]. Accordingly, various point mutations in the *Pparγ* gene, which result in familial partial lipodystrophy are accompanied by severe insulin resistance [395,396,397,398,399]. Consistently, dominant–negative mutations in human *Pparγ* causes metabolic syndrome, insulin resistance, and diabetes at a young age [400,401]. Further, the TZD Pioglitazone impacts the insulin signaling pathway to improve insulin sensitivity [402]. In fact, PPARγ increases the expression of factors in the insulin signaling pathway, including IRS–1 [403], IRS–2 [404], the p85 subunit of PI3K [405], and Cbl–associated protein (CAP) [406,407]. In muscle cells and adipocytes, PPARγ activation increases the expression and translocation of GLUT1, GLUT2, and GLUT4 to the cell membrane stimulating glucose uptake and lowering plasma glucose concentrations [408,409,410]. Further, PPARγ manages the intercellular glucose levels by regulating the expression of genes connected with insulin–stimulated glucose disposal [403,404,405,406,407]. The insulin–sensitizing effect of PPARγ ligands relies on a reduction in local and systemic cytokine production [411] and counteracting tumor necrosis factor α (TNFα) effects [412]. Moreover, PPARγ enhances the plasma levels of adipocytokines, among them adiponectin, which improves insulin sensitivity together with increased free FA oxidation and reduced gluconeogenesis [413,414]. In hepatocytes, Akt2 a downstream factor of insulin signaling stimulates the expression and activity of PPARγ leading to stimulation of glycolysis and lipogenesis [415]. In addition, an interaction between FOXO1 and PPARγ affects insulin sensitivity in adipocytes. FOXO is a transcriptional repressor of *Pparγ*. It binds to its promoter and may also reduce the transcriptional activity of PPARγ by direct protein–protein interaction [416,417,418].

Importantly, the application of PPAR*γ* antagonists can also improve the metabolic profile [419,420], whereas *Pparγ^+/−^* haplodeficiency promotes insulin sensitivity compared with WT animals [421,422]. Similarly, the genetic variants Pro(12)Ala (heterozygotes) and Ala(12)Ala (homozygotes) with reduced transcriptional activity cause leanness and improved insulin sensitivity [377,423,424]. Therefore, a carefully calibrated dose of PPARγ activity is required for optimal insulin sensitivity [425].

The inhibition of mTORC1 disturbs adipogenesis and adipocyte maintenance *in vitro* [120,426,427,428,429], at least partly via PPARγ [120,121,430,431]. Consistently, the application of rapamycin reduces coactivation of some transcription factors (see above) which control lipid metabolism, including PPARγ and PGC–1α [432,433,434]. mTORC1 may activate PPARγ through SREBP–1, which produces endogenous PPARγ ligands [435,436] as well as regulates the hypolipidemic and lipogenic effects of PPARγ [431]. Reciprocally, PPARγ stimulates mTORC1, TG–derived FA uptake, lipoprotein lipase activity, and consequent accumulation of lipids in subcutaneous WAT and BAT. Impact of this pathway by chronic mTOR inhibition results in hyperlipidemia. Rapamycin resistance can be obtained by PPARγ activation, whereas treatment with rapamycin increases both expression and activation of PPARγ in mouse models of breast cancer [437].

PPARγ, similarly to what we discussed for PPARα, also controls autophagy, which is the major cellular process associated with mTOR. PPARγ activation increases autophagy, but its inactivation lessens it [438,439]. Cell cycle arrest and stimulation of autophagy are two major processes by which PPARγ stops cancer cell progression, as evinced in the colon [440], breast [441,442], bladder [443], and adrenocortical [444] cancer cell lines. Contrariwise, in neurons [445,446] and macrophages [447] PPARγ shows anti–autophagic properties.

PPARγ agonists activate AMPK in many cell lines [448,449,450,451,452], tissues ex vivo [453,454], in animals [455,456,457], and humans [458] and improve glucose and fat management. TZDs increase glucose uptake in the liver, adipose tissue, skeletal, and cardiac muscle [454,457,459] as well as reduce insulin resistance via AMPK activation [448,453,460,461]. Agonists of PPARγ stimulate AMPK to enhance expression of adiponectin receptors [458], increase FA oxidation and insulin–stimulated glucose disposal [448], β–cell metabolism [453], and insulin secretion [461]. In addition to metabolic functions, AMPK facilitates the anti–inflammatory activities of PPARγ [451,452]. Reciprocally, while constitutively active or dominant–negative AMPKα reduces PPARγ activity similarly to AICAR and metformin, compound C, an AMPK inhibitor enhances basal and rosiglitazone–stimulated PPARγ activity [462].

PPARγ and SIRT1 functionally cooperate in that PPARγ interacts with SIRT1 to inhibit its activity and by binding to the *Sirt1* promoter to repress its activity [463]. Furthermore, pioglitazone blocks NF–κB activation through the AMPK–SIRT1/p300 pathway [464], and SIRT1 inhibits PPARγ activity by docking with the PPARγ corepressors NcoR and SMRT [465,466]. Consequently, SIRT1 counters adipogenesis, stimulates lipolysis, and the release of fat from mature adipocytes [465,467]. During fasting, SIRT1 represses PPAR*γ* and thereby stimulates fat mobilization [465]. Accordingly, the mobilization of FAs from adipose tissue during fasting is altered in SIRT1^+/−^ mice [465].

#### 6.3.2. PPARγ in Circadian Rhythm

As an important factor in response to CR and metabolic processes, PPARγ is inevitably implicated in circadian rhythm regulation (Figure 3). *Bmal1* [468], *Rev–Erbα* [469], and *Rorγ* [470] are induced by PPARγ. 15–deoxy–Delta12,14–prostaglandin J2 (15d–PGJ2), a natural ligand of PPARγ, promotes the rhythmic expression of endogenous clock genes in NIH3T3 cells. Moreover, 15d–PGJ2 transiently induces *Cry1*, *Cry2*, and *Ror*α expression. However, the 15d–PGJ2–induced entrainment signaling pathway seems to be PPARγ-independent [471]. Similarly, PGC–1α, which interacts with PPARγ, is rhythmically expressed in mouse liver and muscle and it stimulates the expression of *Bmal1* and *Rev–erbα* [169]. PGC–1*α* KO mice are affected in their locomotor activity and are characterized by an altered daily oscillation of body temperature and energy metabolism, in association with the abnormal expression pattern of metabolic and clock genes [169].

In the whole–body *Pparγ* deletion induced in the adult mouse (EsrCre/flox/TM mouse), circadian variations in oxygen consumption, CO_2_ production, food and water intake, locomotor activity, and cardiovascular parameters are repressed and the rhythmicity of the canonical clock genes in adipose tissues and liver, but not skeletal muscles or the kidney, are impaired indicating disturbed circadian rhythms at both behavioral and cellular levels [472]. *Pparγ* shows circadian expression patterns in mouse fat, blood vessels, and liver [288,468]. The *Pparγ* expression in the aorta is robustly rhythmic with a more than 20-fold change during the day/night cycle [468]. *Pparγ* Tie2Cre/flox and SM22Cre/flox KO mice showing *Pparγ* deletion in the endothelial and vascular smooth muscle cells respectively, are characterized by reduced circadian oscillations in heart rate and blood pressure, which is associated with diminished changes in urinary norepinephrine and epinephrine excretion. In these mice, the rhythmicity of the canonical clock genes, including *Bmal1* is compromised [468]. In addition, pioglitazone has been shown to restore the nocturnal blood pressure decline and transform blood pressure from a nondipper to a dipper type in type 2 diabetic patients [473].

Similarly, like PPARγ, the circadian clock is important for adipose tissue functions and fat metabolism. The core clock proteins regulate all aspects of lipid management by cyclic repression of several genes, including those coding for rate–limiting lipolytic enzymes. Consequently, the disruption of the circadian clock causes aberrant dietary lipid absorption, lipid transport, fat storage, and triglyceride levels [474]. The expression of *Rev–Erbα* is stimulated by PPAR*γ* during adipogenesis in rat perirenal and epididymal adipose tissues, as well as in 3T3–L1 adipocytes. REV–ERBα facilitates the expression of PPARγ target genes and, accordingly, ectopic expression of *Rev–Erbα* in 3T3–L1 cells potentiates rosiglitazone PPAR*γ* agonist–induced adipocyte differentiation [469,475]. Therefore, double KO of *Rev–erbα and Rev–erbβ* causes deregulated lipid metabolism [476]. Similarly, *Rorγ* is induced during adipocyte differentiation following the induction of *Pparγ* in D1 and 3T3–L1 cells and functions as an active transcription factor [470]. Also, BMAL1 via WNT signaling controls circadian regulation of adipocyte differentiation [477,478] and embryonic fibroblasts from *Bmal1* KO mice fail to differentiate into adipocytes, whereas over–expression of *Bmal1* in adipocytes increases lipogenesis [477]. Nocturnin, which is controlled by the clock, manages the circadian dietary fat uptake and clock KO mice exhibit altered dietary lipid trafficking [479], whereas PPARγ also plays a role in the uptake of long–chain FA in the intestine [360]. Furthermore, CR increases the amplitude of the circadian clock gene timeless (*Tim*) cycling, which has been implicated in daily fluctuations of several medium–chain triglycerides under CR. Overexpression of *Tim* in peripheral tissues improves oscillatory amplitude, enhances fat metabolism, which extends lifespan under *ad libitum* conditions [480].

HFD induces robustly the hepatic daily expression pattern of *Pparγ*. In HFD–fed mice, the levels of PPARγ protein are elevated, but do not show daily oscillations. Interestingly, however, nuclear and chromatin–bound PPARγ shows a significant circadian oscillation in these mice. Accordingly, PPARγ targets gain rhythmicity after acute HFD feeding [481]. Furthermore, the HFD–related hepatic fluctuations of PPARγ activity are driven by microbial communities in the gut as documented by antibiotic treatments that repress PPARγ-driven transcription in the liver. Therefore, HFD-induced remodeling in the gut microbiota mediates PPARγ-driven modulation of the host liver clock, resulting in adjusted circadian transcription [482].

### 6.4. Systemic Impacts of CR in the Context of Circadian Rhythm and PPAR Activities

#### 6.4.1. Modulation of the Oxidative Status

CR exerts major beneficial effects through reducing ROS levels [483,484,485] in three distinct ways: (i) reduction of oxygen free–radical production by slowing metabolism, (ii) increasing rate of ROS neutralization, and (iii) accelerating the repair of ROS–damaged molecules [486,487,488,489,490]. In vivo, concentrations of ROS and several products of oxidation oscillate in a circadian manner in tissues and blood. The circadian clock is a major regulator of ROS homeostasis as proven by the fact that circadian protein mutants present increased levels of ROS and oxidative damage [491,492] and continuous light diminishes or abolishes the circadian rhythms of SOD, glutathione reductase (GR), and catalase (CAT) [493]. CAT activity peaks in humans at the beginning of the light phase in plasma [494], whereas in mice it culminates in the middle of the dark phase in the liver and kidneys [495] in accordance with eating patterns of these species. The antioxidant proteins peroxidoxins (PRX) show distinct daily fluctuation patterns in the SCN and the liver of Syrian hamsters [493] and, correspondingly, their day time expression changes are accompanied by differences in DNA damage, lipid peroxidation, and protein oxidation [493,496,497,498,499]. In the rat cerebral cortex, SOD activity peaks in the dark phase, correlating with the highest level of malondialdehyde, which is build up in the process of lipid peroxidation [500]. In addition to the cerebellum, both Cu/Zn and Mn–SOD expression levels show diurnal rhythmicity in the rat intestine and lung [501]. These fluctuations may depend on *Per2* since *Per2* KO mice have a lower amplitude of oscillations in hepatic Cu/Zn SOD expression and activity [502]. Along a similar line, KO of both *Per1* and *Per2* results in a 4h shift in the peak of Cu/Zn SOD expression, when compared to WT mice [502].

Reduced glutathione (GSH), in contrast to oxidized GSH (GSSG), is one of the most important non–enzymatic antioxidants. GSH acts as a scavenger of free radicals, supports the reduction of H_2_O_2_, and takes part in detoxification. Circadian oscillations have been demonstrated for GSH and the expression of enzymes necessary for the functional GSH system, including those involved in GSH reduction (GSH reductase (GR)], oxidation of GSH by reduction of H_2_O_2_ [GSH peroxidase (GPx)], and conjugation of GSH [GSH S–transferase (GST)] to various electrophilic substrates [499,503,504,505,506,507,508]. The reduction of GSSG to GSH is necessary to maintain the cellular redox status. This process involves NADP^+^ and its redox partner NADPH, which were mentioned previously as they are inseparably connected with SIRTs. Moreover, in cultured rat fibroblasts high NAD^+^ and NADP^+^ levels decrease the binding capacity of CLOCK:BMAL1 heterodimers to E–box targets, whereas high levels of the reduced NADH and NADPH strongly enhances their binding in concert with BMAL1 acetylation, leading to increased expression of *Pers*, *Crys*, and other circadian genes [202]. Additionally, SIRT1 is also connected with another important marker of the circadian rhythm, melatonin. Melatonin acts as a synchronizer in mammals and provides temporal feedback to oscillators within the SCN and has been proven to display antioxidant properties [509]. Moreover, melatonin inhibits the expression and activity of SIRT1 [510]. Therefore, there is a complex connection between CR and circadian rhythm involving the redox status, SIRT1, and melatonin, at the border of oxidative and circadian systems.

In *Pparα* KO mice, increased oxidative stress occurs at an earlier age compared to WT animals [226] and dosing PPARα agonists restores the redox balance in aged mice [226]. WY–14643 and fenofibrate provide protections against acetaminophen–induced hepatotoxicity by upregulating UCP–2, thereby reducing mitochondrial ROS production [511]. In a gentamicin–induced model of oxidative stress, PPARα and PPARγ agonists (fenofibrate, pioglitazone, tesaglitazar) prevents toxicity by elevating the expression of *Sod*–*1*, glutathione peroxidase 1 and 3 (*GPx1/3*), *Cat*, and uncoupling protein 2 *(Ucp2*) [512].

PPARγ also regulates the expression of several antioxidant and pro–oxidant enzymes, as well as oxidative stress–related proteins, including GPx3 [513], CAT [514,515], and MnSOD [516]. Therefore, heart–specific PPARγ KO mice exhibit extensive oxidative damage accompanied by lower levels of MnSOD and elevated O_2_^−^ levels in cardiac muscle [516]. Furthermore, PPARγ reduces the expression of inducible nitrogen monoxide (NO) synthase (iNOS) and stimulates endothelial NO synthase (eNOS) [517,518,519,520,521]. Conversely, the aorta of endothelial–specific PPARγ KO mice releases less NO than in WT animals, leading to an increase in oxidative stress [519]. Further, activation of PPARγ upregulates UCP2 expression in rats [522], which prevents O_2_^−^ accumulation in the mitochondria and aids in the export of ROS from mitochondrial to the cytosol [523]. Moreover, CD36, the main target gene of PPARγ, supports the recognition and internalization of oxidized lipids [524,525,526]. Finally, PPARγ also increases Bcl–2 levels and thus, protects cardiomyocytes and glial cells from apoptosis triggered by oxidative stress [527,528].

Nuclear factor erythroid 2–related factor 2 (NRF2), a redox–sensitive transcriptional factor that serves as the master regulator of the oxidative stress response, cryoprotection against oxidative as well as electrophilic stress and takes part in inflammation suppression [529] contains PPREs in its gene promoter [530,531]. Reciprocally, NRF2 also binds the upstream promoter region and induces PPARγ [530,532] creating a positive feedback loop between PPARγ and NRF2. Interestingly, BMAL1 directly controls the expression of NRF2 [533], Hence, there is an overlap between the ability of PPARs to control oxidative stress and the effects of CR and circadian rhythm (Figure 4).

#### 6.4.2. Reduction of Inflammation

The immune system as one of the key factors indispensable for survival has been intensively investigated in the context of circadian rhythms. The expression levels of toll–like receptors (TLRs) display significant circadian fluctuations in the mouse jejunum [534]. In the spleen and natural killer cells of rats, transcription of interferon γ (*Ifnγ)*, granzyme B (*GrB*), perforin (*Prf*), and *Tnfα* oscillate showing the highest expression at the break of the dark and the light phase which coincides with the cytolytic activity of splenic natural killer cells [535,536,537]. Similarly, the migration of lymphocytes through lymph nodes is regulated by circadian rhythm and peaks at the beginning of the active phase in mice [538]. Accordingly, the timing of triggering and generating immune response is decisive in the regulation of the efficiency of the reaction. After infection of the mouse gastrointestinal tract with *Salmonella enterica serovar Typhimurium* there is a more efficient bacteria clearance for infection done 4 h after the beginning of the dark phase compared to 4h into the light phase. Moreover, levels of antimicrobial peptide lipocalin–2 in the intestine are higher during the day than during the night [539]. Importantly, PPARγ and lipocalin–2 cooperate in response to bacterial infection in the intestine [540]. In the case of sepsis, the timing of the disease induction impacts the severity of the disease and coincides with the oscillations in TLR9 expression [541].

The central pacemaker can also regulate oscillations of inflammatory factors in peripheral organs as local splenic sympathectomy alters daily variations of cytokines, TNFα, and cytolytic factors, GRB, and PRF, in natural killer cells and splenocytes [542]. A lesion in the SCN leads to loss of daily variations in serum histamine, MCP–1 (CCL2), and IL–6 levels in mice [543] and dysregulated immune response with higher levels of cytokines after administration of lipopolysaccharide (LPS) in rats [544]. Moreover, night administered LPS increases SCN basal neuronal activity [544] and induces more severe responses in body temperature and cytokine production than LPS given in the rest period [544]. However, the inflammatory response of the mouse lung to LPS is strongest during the day compared to the night [545]. Finally, rodents with a reversed feeding schedule, consuming during the light phase, and fasting at night, show inadequate inflammatory response to LPS in association with elevated TNFα and IL–6 levels and increased mortality [546,547].

Clock genes have been shown to modulate the expression of multiple inflammatory cytokines. CRY affects TNF*α*, IL1β, and Il–6 [548], ROR*α* impacts Il–1β, and Il–6 [548], PER2 modulates daily rhythms of IFNγ [549], and Il–1β [550], Rev–ERB*α* regulates IL–6 [551], MCP-1 as well as TH17 cells [552], while Bmal1 regulates diurnal oscillation of Ly6C^hi^ inflammatory monocytes [553] and reduces monocyte recruitment [554]. Consequently, *Clock* KO mice exhibit reduced cytokine production in macrophages in response to the LPS challenge or *Salmonella Typhimurium* infection [539]. On the contrary, deletion of the negative clock arm *Ror–α* in mice leads to a stronger LPS–induced inflammation with higher levels of Il–1β, IL–6, and macrophage inflammatory protein–2 in bronchoalveolar lavage fluids from mice lungs [548]. Moreover, T and B cell development is defective in these mice while mast cells and macrophages produce an increased amount of TNF*α* and IL–6 upon activation [555]. *Per2* KO mice are more resistant to the LPS challenge compared to WT mice [550]. KO or silencing of *Cry1* and *Cry2* also leads to increased inflammation, a permanent elevation of proinflammatory cytokines in a cell–autonomous manner, and constitutive activation of NF–κB and PKA signaling [556]. Importantly, CLOCK is required for acetylation of p65, a key event for NF–κB transactivation and downstream cytokine production [557]. NF–κB is a transcriptional regulator of major importance and its action is usually pro–inflammatory and pro–oxidant. NF–κB induces the expression of the inflammatory cytokines sIL–1β, IL–6, and TNFα, as well as the proinflammatory enzymes cyclooxygenase–2 (COX–2) and iNOS. However, it can also affect the expression of *Sod* and other anti–inflammatory genes [558,559,560,561,562,563]. CR exhibits a broad anti–inflammatory effect and it acts, among others, through NF–κB and I*κ*B to blunt age–triggered increases in COX–2 levels and activity. CR suppresses the generation of iNOS, IL–β, IL–6, TNFα, and prostanoids, such as TXA 2, prostacyclin 2, and prostaglandin E2 [59,485] while dampening aging–related reduction of PPAR expression and activity [229]. PPARs interfere with several pro–inflammatory mediators including NF–*κ*B, signal transducer and activator of transcription 1 (STAT1), and activating protein–1 (AP–1) [564,565,566,567] inhibiting the expression of genes coding for inflammatory proteins, such as COX–2, iNOS, cytokines, metalloproteases, and acute–phase proteins [520,565]. Moreover, inflammatory eicosanoids including prostaglandins and leukotrienes act as ligands for PPARs [568,569]. However, each PPAR exhibits a set of distinct anti–inflammatory activities [569].

The anti–inflammatory properties of PPAR*α* are mostly based on its interaction with NF–*κ*B. Accordingly, PPAR*α* KO results in premature oxidative stress and an aggravated inflammatory response as well as a stronger age–dependent increase in oxidative stress and NF–*κ*B activity [226,570]. In agreement with these observations, agonists of PPARα reduce the activity of NF–κB and the spontaneous generation of inflammatory cytokines [226,571]. Importantly, high doses of PPAR*α* ligands induce NF–*κ*B, whereas low amounts reduce NF–*κ*B activation [226]. PPARα contributes to the anti–inflammatory properties of CR by reducing the expression of acute–phase genes (*C4bp*, *C9*, *Mbl1*, *Orm1*, and *Saa4*) responsive to inflammatory cytokines [232].

PPARβ/δ is also capable of regulating NF–*κ*B. The activation of PPARβ/δ protects skeletal muscle against metabolic disorders also by means of counteracting the diet–induced activation of NF–*κ*B as well as expression of iNOS and intercellular–adhesion–molecule–1 (ICAM–1) [572]. GW0742, an agonist of PPARβ/δ, inhibits LPS–triggered TNF*α* secretion. Accordingly, overexpression of PPARβ/δ substantially inhibits TNF*α* expression, while the absence of PPARβ/δ enhances LPS–induced TNF*α* production in cultured cardiomyocytes [573]. In the intestine, PPARβ/δ dampens inflammatory signaling and therefore it can suppress inflammatory bowel disease [574].

PPARγ is one of the most potent anti–inflammatory factors with extensive research proving its properties. PPARγ agonists diminish inflammatory bowel disease symptoms and are instrumental in the therapy of ulcerative colitis and Crohn’s disease [540,575,576,577,578,579,580]. The ligands of PPARγ prevent macrophage activation, promote macrophage conversion into the non–inflammatory type M2, and inhibit the production of inflammatory cytokines in macrophages and dendritic cells. Consequently, PPARγ KO mice show increased susceptibility to infection [520,565,581,582] whereas, mice specifically deficient in colonic PPARγ exhibit more acute symptoms of infectious colitis [540] and are resistant to colitis therapy using conjugated linoleic acid [583]. *Pparγ* polymorphism is associated with changes in the onset of diseases with an inflammatory background including multiple sclerosis [584], T2D [424], vascular morbidity, and mortality [585], as well as colorectal cancer risk [586,587]. PPARγ exerts its anti–inflammatory properties by inhibiting the expression of inflammatory genes encoding cytokines, metalloproteases, and acute–phase proteins, and affecting multiple signaling molecules including p53 [588], Bcl2 [356], c–Myc, [589], Cox–2 [590,591], iNOS [592], and Apc/β–catenin [593,594]. Most relevantly, PPARγ inhibits NF–κB, and NF–κB–driven transcription [356,591]. Contrariwise, NF–κB reduces the transcriptional activity of PPARγ via a mechanism that involves HDAC3 [595,596]. Therefore, NF–κB seems to be a universal inflammatory factor connecting circadian rhythm, CR, and all of the PPARs (Figure 4).

#### 6.4.3. Prevention of Aging

Circadian clocks have been implicated in the control of aging and aging–associated pathologies via regulation of oxidative stress, cell cycle, cell death, and DNA damage response [597]. Importantly, aging is accompanied by a shift in the phase and a decrease in amplitude of rhythmic oscillations [598,599,600,601]. In mice, age–dependent degradation of neuronal activity rhythms in the SCN results in the disruption of the physiological ability to entrain to external light cues, and approximately 20 times reduced sensitivity to the synchronizing effect of light compared to young animals; this leads to an alteration of rhythmic behavior [602,603,604,605,606]. Circadian rhythm–related changes during aging include body temperature, activity–wakefulness, locomotor activity patterns, and drinking behavior. In humans, aging is linked with a phase advance in the rhythms of body temperature and melatonin secretion as well as the earlier habitual time of sleeping and disturbed sleep [607,608]. Further, age alters the 24 h expression profile of *Clock* and *Bmal1* in the hamster SCN [609] and results in alterations in the circadian transcriptome. The decline occurs in a tissue–specific manner, as documented by the comparison of transcriptional patterns in the liver versus epidermal and skeletal muscle stem cells [70].

*Bmal1* KO mice have reduced lifespans and display various symptoms of premature aging including sarcopenia, cataracts, lesser subcutaneous fat, and organ shrinkage [13]. Deficiency of CLOCK also impacts longevity with the average and maximum lifespan of *Clock KO* mice being reduced by 15% and 20%, respectively, compared to WT mice. This phenotype is accompanied by a high rate of development of age–related pathologies, cataracts, and dermatitis [11]. However, the mice do not develop the premature aging phenotype that characterizes *Bmal1* KO mice. Based on wheel–running activity, the circadian rhythm in *Per1, 2* KO mice is out of control. These animals also display signs of premature aging including loss of soft tissues and kyphosis, a more rapid decline in fertility and litter size in each successive breeding, as well as a significant increase in neoplastic and hyperplastic phenotypes [610]. *Per1* and *Per2* were proposed as longevity–associated candidate genes based on (i) the KO phenotype in mice, (ii) comparison of the activity of a group of genes with differential expression in several tissues such as the liver, brain, skeletal muscles of mice subjected to CR and (iii) *Per* gene expression in the long–lived dwarf mice [610,611]. In golden hamsters carrying a 20 h–period mutation, *tau*, resulting in 24 h light–dark cycles, life span is shortened. However, in aged animals, longevity can be extended by fetal suprachiasmatic implants which restore higher amplitude rhythms [612,613,614].

Oscillation disturbance by weekly reversal of the light–dark cycle reduces the survival time of cardiomyopathic hamsters [615]. Transplanting fetal SCN to an old hamster with weak behavioral rhythms restores robust rhythmicity in the transplanted hosts [616], triggers a greater response to the phase–shifting effects of triazolam [617], and increases lifespan by 4 months [612]. In humans, a disrupted circadian rhythm in night–shift workers leads to metabolic disorders, increased incidence of cancer, hormone imbalance, gastrointestinal abnormalities, cardiovascular diseases, reproductive aberrations, sleep and psychological disorders including depression and anxiety [618].

In young mice, 2626 liver genes present a rhythmic expression, while in old mice this number is reduced to 1664 genes, including 1575 genes that are rhythmic in both groups and comprise the core clock genes. CR increases the number and amplitude of rhythmic genes in both young and old mouse livers [70]. Besides gene expression, also global protein acetylation is highly circadian. The acetylation rhythm is dampened in aged mice and can be rescued by CR [70]. Moreover, in aging epidermal stem cells (epSCs) and skeletal muscle stem cells (muSCs), CR maintains or restores rhythmic homeostatic functions [619]. As in mice, CR in *Drosophila* enhances the amplitude of cycling of most circadian clock genes, including *Tim*, in peripheral tissues [480]. *Per* and *Tim* KOs attenuate the CR–dependent changes in lifespan extension and fat metabolism [480]. Similarly, 30% CR does not increase the lifespan or have an impact on the physiology of *Bmal1* KO mice [71]. Therefore, some of the circadian clock genes are necessary for the full benefits of CR.

Aging is associated with decreased metabolic rate, impaired mitochondrial function, reduced insulin resistance, dysfunctional lipid metabolism, immunosenescence, increased oxidative stress, and lowered hormonal secretion [620,621,622]. Glucose metabolism and insulin sensitivity maintenance is a key feature of the anti–aging actions of CR [54,75]. In fact, genes connected with the insulin/IGF–1 signaling pathway, which include PPARs, have been proposed as longevity candidate markers [77,78,623]. Similarly, healthy adipose tissue has also been directly associated with lifespan due to its role in energy metabolism [74,391], which strongly relies on PPARγ [624,625,626]. Therefore, in gene network analysis, PPARγ was identified as one of the “longevity genes” in mouse WAT [627].

Most of the multiple traits of aging are related to PPARs and mTOR activity, oxidative stress, inflammation, and metabolism (Figure 4). Furthermore, aging–associated changes in PPAR expression and activity are reversed by CR [229,268,628,629]. The impact of PPARs can be particularly well observed in mutant models of longevity, the dwarf mice, a model that shows multiple similarities with CR animals [630]. These mice are characterized by a hepatic increase in PPAR*α* expression and constitutive activation of some of its target genes including genes involved in β– and ω–oxidation of FAs [220,631]. Growth hormone receptor (GHR) KO dwarf mice show similar expression patterns of PPAR*α* in the muscle and PPARβ/δ in the liver and skeletal muscle as WT CR mice [225]. Fittingly, mice overexpressing the bovine GH gene, which are characterized by short life span, exhibit a decrease in hepatic expression of PPARα and genes involved in FA activation, β–oxidation as well as the production of ketone bodies leading to limited capacity to adjust to fasting [632]. Furthermore, overexpression of FGF21, a PPAR*α* target gene, extends lifespan in mice and blunts GH/IGF–1 signaling in the liver [633]. Therefore, the resetting of circadian rhythms by CR can lead to increased longevity and PPARs play a vital role in the rejuvenating outcomes of CR.

## 7. Conclusions

Importantly, not only feeding schedule or feeding restriction (CR) but also nutrients entrain the circadian rhythm. A balanced combination of carbohydrate and protein induces phase advance in the liver clock, whereas the application of a single macronutrient (protein, carbohydrate, or fat) had no significant impact [634]. Similarly, the liver clock can be reset by the ingestion of a mix of glucose and amino acids, but not by either of the nutrients alone [26]. Therefore, various types of diet–related cues trigger an adjustment of circadian oscillations to adapt to the current metabolic status. In the case of CR for example, tissues need to optimize and synchronize their metabolic activity to reach beneficial homeostasis. The circadian clock controls the expression and activity of many rate–limiting enzymes and fine–tunes mechanisms of metabolic optimization and energy homeostasis. Because of its capacity to enable anticipation of daily events, it is superior in directing efficient use of energy. Resetting the circadian clocks through light, feeding, or pharmacological intervention grants metabolic health and rejuvenation underscoring the importance of the interplay between diets and biological clocks to promote longevity. Targeting molecular executors at the crossroad of CR and the circadian clock, like PPARs, may be an efficient and promising way to take advantage of the benefits of a well-calibrated circadian energy metabolism.

## Figures and Tables

**Figure 1 nutrients-12-03476-f001:**
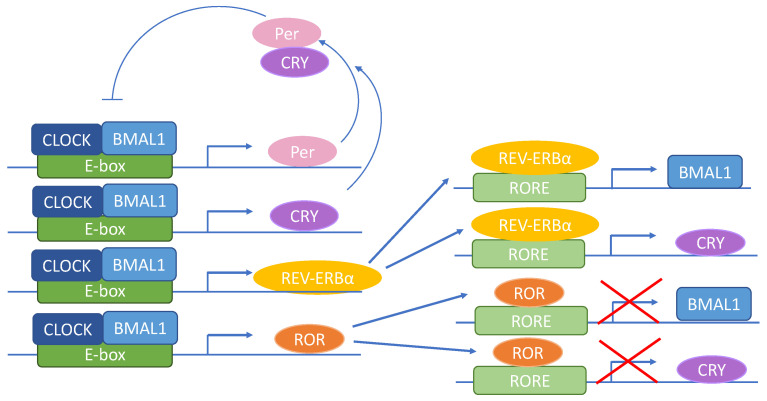
Transcription feedback loops driving circadian rhythm. CLOCK and BMAL1 heterodimers initiate expression of PERs, CRYs, REV–ERBα, and RORs by binding to the E–box element. PER and CRY heterodimers inactivate CLOCK and BMAL1. REV–ERBα and RORs activate or inhibit, respectively, the expression of ROR response element (RORE) containing genes including *Cry* and *Bmal1*. Abbreviations: BMAL1—brain and muscle ARNT–like protein 1; CLOCK—circadian locomotor output cycles kaput; CRY—cryptochrome; PER—period, ROR—retinoic acid–related orphan receptor.

**Figure 2 nutrients-12-03476-f002:**
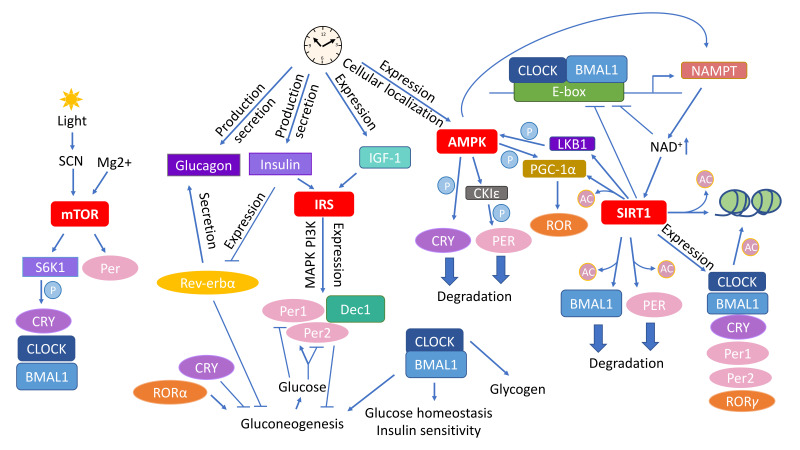
Circadian rhythm affects the main molecular pathways mediating the outcome of caloric restriction (CR). Various types of interactions connect the CR–associated pathways mTOR, insulin signaling, AMPK, and sirtuins with circadian rhythm. Abbreviations: AMPK—adenosine monophosphate (AMP) activated protein kinase; CKIε—casein kinase ε; CLOCK—circadian locomotor output cycles kaput; Dec1—differentially expressed In chondrocytes 1; IGF–1—insulin–like growth factor 1; IRS—insulin receptor substrates; LKB1—liver kinase B1; mTOR—mammalian target of rapamycin; NAD—nicotinamide adenine dinucleotide; NAMPT—nicotinamide phosphoribosyltransferase; SIRT1—Sirtuin 1; SCN—suprachiasmatic nucleus, S6K1—ribosomal protein S6 kinase 1. The circled Ac indicates acetylation and circled P phosphorylation.

**Figure 3 nutrients-12-03476-f003:**
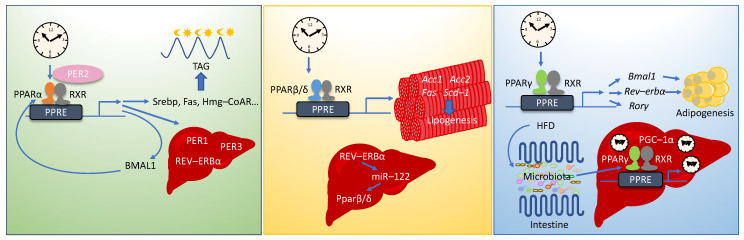
The role of PPARs in circadian rhythm. The levels of all PPARs are regulated according to daily oscillations. PPARα targets the expression of lipid metabolism–related genes resulting in rhythmic changes in the levels of triacylglycerols (TAG). PPARα interacts directly with PER2 to modulate gene expression. In the muscle, PPARβ/δ regulates the expression of lipogenic genes in a rhythmic manner. PPARβ/δ is also an indirect target of the circadian clock via miR–122. PPARγ regulates expression of core circadian genes including *Bmal1* and *Rev*–*erbα*, which support adipogenesis. In the liver, PPARγ also mediates the action of circadian triggers derived from gut microbiota in response to high–fat diet (HFD). PPARs bind to PPRE as PPAR:RXR heterodimers. Abbreviations: Acc—acetyl–CoA carboxylase; Fas—fatty acid synthase; Hmg–CoAR—β–hydroxy β–methylglutaryl–CoA reductase; PGC–1α—peroxisome proliferator–activated receptor γ coactivator 1 α; PPRE*—*PPAR response element; RXR—retinoid X receptor; Scd–1—stearoyl–CoA desaturase–1; Srebp—sterol regulatory element–binding protein 1.

**Figure 4 nutrients-12-03476-f004:**
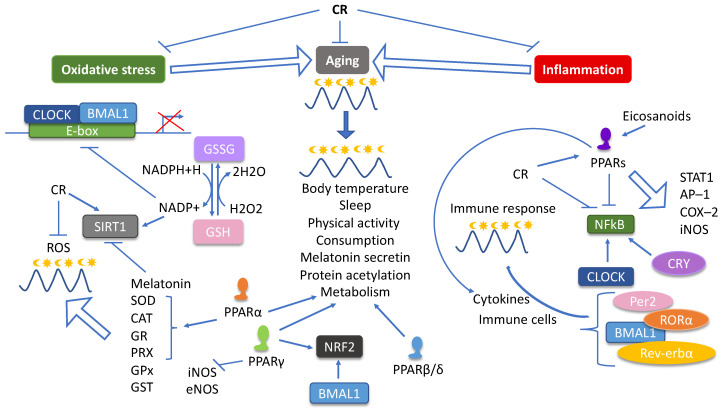
PPARs contribute to the processes regulated by CR and circadian rhythm at the whole–body level. CR exerts its beneficial effects by reducing oxidative stress, delaying aging, and inhibiting inflammation. Circadian rhythm affects all of these processes. PPARs regulate the expression of antioxidative enzymes, inflammatory cytokines, and metabolic processes. For the effects of SIRT refer to Figure 2. Abbreviations: AP–1—activating protein–1; CAT—catalase; COX–2—cyclooxygenase–2; eNOS—endothelial NO synthase; GPx—GSH peroxidase; GR—GSH reductase; GSH—glutathione; GSSG—oxidized glutathione; GST—GSH S–transferase; iNOS—inducible NO synthase; NRF2—nuclear factor erythroid 2–related factor 2; PRX—peroxiredoxins; ROS—reactive oxygen species; SOD—superoxide dismutase; STAT1—signal transducer and activator of transcription 1.

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
