# Peer review of "Peroxisome Proliferator-Activated Receptors as Molecular Links between Caloric Restriction and Circadian Rhythm"

_nutrients, 2020, doi:10.3390/nu12113476_

Round 1

Reviewer 1 Report

Dear editor,

The present work is a very comprehensive and detailed review of the relationship between circadian rhythm, caloric restriction and the role of PPARs in this. A well-documented and extensive overview of the molecular mechanisms and pathways involved is provided. Overall, the quality and quantity of the information provided is excellent.

I have only 2 main observations:

One missing point is the regulation of food intake by PPARs at the CNS (e.g. 10.1016/j.physbeh.2016.04.011; 10.1016/j.cmet.2016.12.006; 10.1016/j.metabol.2018.06.005; 10.1172/JCI74915), which has been studied previously by the same authors, and therefore must be included.

Section 6 first paragraph needs to be reviewed and reworded, the core idea is understandable but after a few reads. The sentences in line 430-434 about the expression of PPARα in the hearth has some contradictory statements.

A few minor points involve mistakes in formatting:

In line 320 GLUT4 is underscored.

In line 399 the first 8 words are in italics.

In line 988-989 there are words in italics that are not gene names

Reference 617 is not included in the text.

Author Response

The present work is a very comprehensive and detailed review of the relationship between circadian rhythm, caloric restriction and the role of PPARs in this. A well-documented and extensive overview of the molecular mechanisms and pathways involved is provided. Overall, the quality and quantity of the information provided is excellent.

We thank the reviewer for this very positive review of our manuscript

I have only 2 main observations:

One missing point is the regulation of food intake by PPARs at the CNS (e.g.10.1016/j.physbeh.2016.04.011; 10.1016/j.cmet.2016.12.006; 10.1016/j.metabol.2018.06.005; 10.1172/JCI74915), which has been studied previously by the same authors, and therefore must be included.

Reply

Thank you for your comment and encouragement to cite our own work. Information concerning each of PPARs in the context of food intake and CNS have been added in sections 6.1., 6.2., and 6.3.

Section 6 first paragraph needs to be reviewed and reworded, the core idea is understandable but after a few reads. The sentences in line 430-434 about the expression of PPARα in the hearth has some contradictory statements.

Reply

Sentence structure in the first paragraph of section 6 has been simplified as requested. The mistake concerning the expression of PPARα in the heart during CR has been corrected.

A few minor points involve mistakes in formatting:

In line 320 GLUT4 is underscored.

In line 399 the first 8 words are in italics.

In line 988-989 there are words in italics that are not gene names

Reference 617 is not included in the text.

Reply

The formatting in the indicated lines has been corrected

Reviewer 2 Report

Thank you for this manuscript which is of great scientific interest, the main regulatory elements of the circadian rhythm are presented, very well documented, with synthesis diagrams.
Only missing, a list of abbreviations mentioned in the manuscript would be included.

Title. 

Include Caloric restriction.

Introduction.

The introduction is well written, please include a non systematic review. 

The manuscript.

All elements are very well described; there is a logic link between the different parts of the manuscript. 

Several diagrams illustrate the parts

Only an important point for me is to include a list of many abbreviations used in this manuscript.

The manuscript is very complete and very interesting to read.

Author Response

Thank you for this manuscript which is of great scientific interest, the main regulatory elements of the circadian rhythm are presented, very well documented, with synthesis diagrams.
Only missing, a list of abbreviations mentioned in the manuscript would be included.

We thank the reviewer for this very positive review of our manuscript

Title. 

Include Caloric restriction.

Reply

The title has been changes accordingly.

Introduction.

The introduction is well written, please include a non systematic review. 

Reply

The phrase “non-systematic review” has been added in the abstract (line 22) and the introduction (line 83)

The manuscript.

All elements are very well described; there is a logic link between the different parts of the manuscript. 

Several diagrams illustrate the parts

Only an important point for me is to include a list of many abbreviations used in this manuscript.

The manuscript is very complete and very interesting to read.

Reply

The list of abbreviations has been added at the end of the manuscript.